# Multilevel factors associated with delays in screening, diagnosis, and treatment for lung cancer—A mixed methods systematic review protocol

**Sabina O. Nduaguba**[1,2]*, **Kimberly M. Kelly**[1,2]

**1** Department of Pharmaceutical Systems and Policy, College of Pharmacy, West Virginia University, Morgantown, West Virginia, United States of America, **2** West Virginia University Cancer Institute, Morgantown, West Virginia, United States of America

* sabina.nduaguba@hsc.wvu.edu

## Abstract

**Data Availability Statement:** No datasets was generated or analysed during the preparation of the protocol.

### Background

Factors affecting time to lung cancer care may occur at multiple levels of influence. Mixed-methods reviews provide an approach for collectively synthesizing both quantitative and qualitative data. Prior reviews on timeliness of lung cancer care have included only either quantitative or qualitative data, been agnostic of the multilevel nature of influencing factors, or focused on a single factor such as gender or socioeconomic inequalities.

### Objective

We aimed to update the literature on systematic reviews and identify multilevel factors associated with delays in lung cancer screening, diagnosis, and treatment.

### Design

The proposed systematic review will be conducted in accordance with the Joanna Briggs Institute (JBI) Manual for Evidence Synthesis specific for mixed methods systematic reviews. Reporting will be consistent with PRISMA guidelines.

### Methods

Medline (PubMed), CINAHL, and SCOPUS will be searched using validated search terms for lung cancer and factors, health disparities and time/delay. Eligible studies will include original articles with quantitative, qualitative, or mixed-methods designs that investigate health disparities in, risk factors for, or barriers to timely screening, confirmatory diagnosis, or treatment among patients with lung cancer or those at risk for lung cancer. Title, abstract, and full-text screening, study quality assessment, and data extraction will be conducted by two reviewers. A convergent integrated approach with thematic synthesis will be applied to synthesize the extracted and generated analytical themes.

**Funding:** Funding for this work was provided to Sabina Nduaguba as a subgrant from the National Cancer Institute (NCI) Geographic Management of Cancer Health Disparities Program (GMaP, http://gmapr1.com/). The grant number is: P30 CA177558-10S1. GMaP played no role in the study design, decision to publish, or preparation of the manuscript.

**Competing interests:** The authors have declared that no competing interests exist.

## Discussion

Findings from this review will inform the design of an intervention to address delays in lung cancer screening for high-risk persons, diagnosis of suspected lung cancer, and treatment of confirmed cases.

## Introduction

Lung cancer is the second most common cancer in both men and women and the leading cause of cancer mortality in the US [1, 2]. Five-year survival for patients with lung cancer is 25% ranging from 7% to 63% for late-stage vs early stage diagnoses [3]. Only 16% of lung cancers are diagnosed early [4], and the wait time to initiation of appropriate treatment following from initial suspicion of lung cancer may be long [5–14]. Delayed treatment, particularly treatment with curative intent, has also been shown to be associated with poor survival [5, 7, 15–19]. Delays occur at multiple stages along the lung cancer care pathway that cumulatively prevent the timely receipt of lifesaving treatment. Intervals of delay include cues to screening, diagnostic work-up following a positive screen, and time taken to initiate treatment. These delays are potentially preventable and can be targets for intervention to improve lung cancer outcomes.

Factors affecting time to lung cancer care occur at multiple, expanding spheres of influence, ranging from the micro or individual level to the macro or societal level. According to Taplin's ecological framework, the levels of influence occur at the individual, family/social support, provider/team, organization/practice setting, and environment which are identified as levels of contextual influence that affect behaviors along the cancer care continuum [20, 21]. These multilevel factors could potentially interact, increasing their influence on timeliness of receiving care. For example, in the US, non-Hispanic Blacks and Hispanics are more likely than non-Hispanic Whites to live below the poverty line [22], and the interaction of race/ethnicity and poverty status has implications for access to and timeliness of receiving cancer care [23, 24]. Furthering the example, racial/ethnic minorities, particularly Blacks, have been shown to exhibit some mistrust of the health system. The interaction of race/ethnicity and medical mistrust, at the health system level, can also lead to delays in receiving cancer care by affecting the quality of communication between providers and their patients with lung cancer [25]. Such interaction among factors requires multilevel interventions (MLIs) to address both proximal determinants of the problem at their various levels of influence and achieve distal outcomes.

Although prior reviews have been conducted on the timeliness of receiving lung cancer care, these had some limitations. In a systematic review by Cassim et al. of qualitative studies on barriers to early diagnosis of lung cancer, identified barriers were classified into disease, patient, and healthcare and system factors [26]. A different review of quantitative studies identified individual (age, comorbidity, and atypical symptoms at presentation in a hospital setting) and health system factors (treatment intent, type of physician at initial referral, number of diagnostic tests, and number of facilities involved) [27]. Other reviews on timeliness of receiving lung cancer care have been either agnostic of the multilevel nature of influencing factors [27–30] or focused on a single factor such as gender [31] or socioeconomic inequalities [32]. They also included studies from different countries and were restricted to only qualitative or quantitative studies, which have implications both for the contextualization of findings across different geographic areas and also for the breadth of synthesized evidence across study designs.

Mixed-methods systematic reviews provide an approach for collectively synthesizing both quantitative and qualitative data. Considering that previous systematic reviews were limited to either only quantitative or qualitative data and/or had an international scope, a mixed-methods systematic review on multilevel factors affecting the timeliness of receiving lung cancer care in the US will provide a timely and comprehensive update. A comprehensive understanding of the contextual factors that influence the timeliness of lung cancer care at the various levels of influence will inform the design of MLIs to reduce time to receipt of care along the care continuum. Such interventions will ultimately improve outcomes for patients with lung cancer.

## Objectives

The objective of this systematic review is to identify multilevel factors associated with delays experienced by patients along the lung cancer care continuum from screening to diagnosis and treatment in the US. To achieve this, we propose to address the following aims:

1. To identify the multilevel factors associated with delays in lung cancer *screening* in the US

2. To identify the multilevel factors associated with delays in lung cancer *diagnosis* in the US

3. To identify the multilevel factors associated with delays in lung cancer *treatment* in the US

## Materials and methods

The proposed systematic review will be conducted in accordance with the Joanna Briggs Institute (JBI) Manual for Evidence Synthesis specific for mixed methods systematic reviews (Fig 1) [33]. Reporting will be conform to the Preferred Reporting Items for Systematic reviews and Meta-Analysis (PRISMA) guidelines [34]. The protocol for the proposed study is registered with PROSPERO (CRD42022346097).

### Outcomes

Our outcome of interest is delay in receipt of lung cancer care and includes: 1) delay in screening for lung cancer; 2) delay in diagnosis of lung cancer; and 3) delay in receipt of lung cancer treatment. Delay is defined based on the time intervals between two points along the cancer care continuum or wait times to receive care and could include non-receipt, refusal or failure to follow-up on provider recommendation for care.

### Informational sources and search strategy

The following databases will be searched from inception for published studies in the literature–Medline (PubMed), CINAHL, and SCOPUS–using validated search terms lung cancer [35] and terms specific for factors [36], health disparities [37] and time/delay [27, 32, 38, 39] adapted from prior reviews for the four databases (Table 1) [40].

### Eligibility criteria

Eligible studies will be original articles with quantitative, qualitative, or mixed-methods designs that investigate health disparities in, risk factors for, or barriers to timely screening, confirmatory diagnosis, or treatment among patients with lung cancer or those at risk for lung cancer. Studies that investigate factors associated with whether or not patients receive lung cancer care will also be included as non-receipt of care is also indicative of delays along the

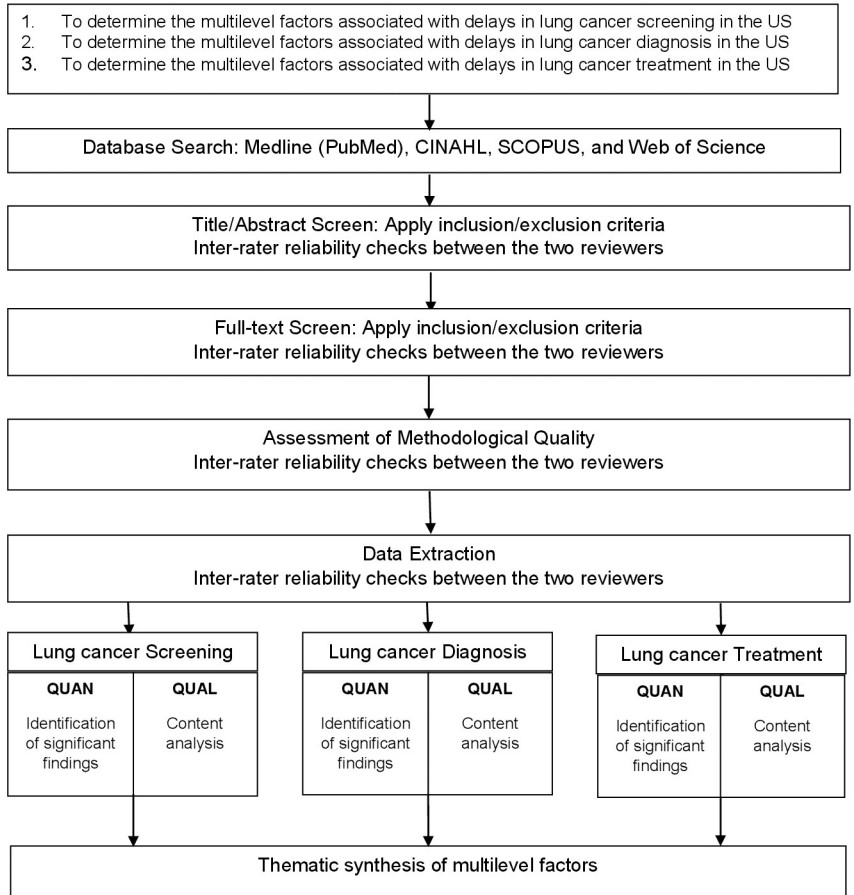

**Fig 1. Mixed-method systematic review design flow chart.** QUAN = Quantitative data; QUAL = Qualitative data.

cancer care continuum. To contextualize synthesized findings within the environment in which the factors affecting lung cancer care are experienced, included studies will be restricted to primary studies conducted in the US with full-text articles published in English. Letters to the editor, editorials, and commentaries will be excluded. The identified study titles and abstracts will be screened against the selection criteria by two independent reviewers for relevance. Selected studies will be retrieved in full and undergo full-text screening by two independent reviewers based on inclusion criteria. Reasons for excluded studies will be documented. Disagreements will be resolved through discussion. In the event that consensus is not reached, input will be sought from a third reviewer to resolve the discrepancy.

## Quality assessment

Methodological quality will be assessed by two independent reviewers using the NIH Quality Assessment Tool for quantitative studies [41] and the Consolidated Criteria for Reporting Qualitative Research checklist for qualitative studies [42]. Disagreements will be resolved through discussion. The NIH Quality Assessment Tool is a collection of tools developed by the National Heart, Lung, and Blood Institute and assesses the internal validity of a study. This tool(s) offers the advantage of assessing multiple studies based on different quantitative research designs including intervention studies, observational cohort and cross-sectional

**Table 1. MEDLINE (PubMed) search strategy.**

| S/N | Search Terms |
|---|---|
| 1 | ((Lung neoplasms[mh:noexp] OR bronchial neoplasms[mh] or pancoast syndrome[mh] or lung cancer*[tw]) AND humans[sb]) OR (NSCLC[ti] OR SCLC[ti] OR (pancoast[ti] AND (syndrome[ti] OR tumor*[ti] OR tumour*[ti]))) OR ((lung*[ti] OR pulmonary[ti] OR bronchus[ti] OR bronchogenic[ti] OR bronchial[ti] OR bronchoalveolar[ti] OR bronchioalveolar[ti] OR bronchioloalveolar[ti]) AND (cancer*[ti] OR carcinoma*[ti] OR adenocarcinoma*[ti] OR malignan*[ti] OR metastas*[ti] OR tumor*[ti] OR tumour*[ti] OR neoplasm*[ti])) NOT medline[sb]) NOT (meta analysis[pt] OR meta analysis[tiab] OR meta analysis[mh] OR review[pt] OR search*[tiab])AND English[la] |
| 2 | Healthcare Disparities/ OR Health Equity/ OR Health Status Disparities/ OR Culturally Competent Care/ OR Social Determinants of Health/ OR Sociology, Medical/ |
| 3 | (disadvantaged[tiab] OR discriminat*[tiab] OR disparat*[tiab] OR disparit*[tiab] OR disproportion*[tiab] OR inequal*[tiab] OR inequit*[tiab] OR unequal[tiab] OR underserved[tiab] OR under-served[tiab] OR (cultural* adj compet*[tiab]) OR (social* adj determin*[tiab])) OR (difference[ti] OR different[ti]) |
| 4 | predict*[ti] OR barrier*[ti] OR challeng*[ti] OR factor[ti] OR factors[ti] OR determine*[ti] OR relationship*[ti] |
| 5 | time factors[MeSH] OR waiting lists[MeSH] OR delay[tw] OR Timeliness[tw] OR Time[tw] OR prognosis[MeSH] |
| 6 | delay*[tiab] OR interval*[tiab] OR time*[tiab] OR pathway*[tiab] OR route*[tiab] OR wait*[tiab] OR timeline[tiab] OR timeliness[tiab] OR timeframe[tiab] OR period[tiab] OR periods[tiab] OR latency[tiab] OR late[tiab] OR lateness[tiab] |
| 7 | #2 OR #3 OR #4 |
| 8 | #5 OR #6 |
| 9 | #1 AND #7 AND #8 |

studies, case-control studies, pre-post studies with no control group, and case series studies. Consistent with guidelines from the NIH Quality Assessment Tool, the overall quality rating will be based on the totality of flaws identified.

## Evidence synthesis

**Data extraction.** Qualitative and quantitative data will be extracted from the full-text of the included studies using JBI data extraction tool adapted for a spreadsheet (S1 File). Article related data will be extracted including study-reference (authors' names, publication year) and context (setting in which study is conducted). Study type and methodology will be extracted, along with participant data: age, gender, LGBTQ+ status, race, ethnicity, socioeconomic status, disability status, rurality, Appalachian status, and number. Phenomena of interest include health disparities (for studies that identify disparate groups that experience delays to lung cancer care), risk factors (for studies that identify predictors of delays in receiving lung cancer care), and barriers (for studies that identify reasons for delays to lung cancer care). Time to lung cancer screening, diagnosis, and treatment as well as risk factors/barriers identified will also be noted. In addition, consistent with the JBI SUMARI tool, qualitative studies will include themes supported by illustrations, along with level of credibility [33]. Disagreements will be resolved through discussion.

**Data transformation.** Results from quantitative data will be transformed into 'qualitized data' [43]. This will involve transformation of quantitative data into textual descriptions using narrative synthesis to describe factors with significant findings ($p < 0.05$) along with the relevant numerical data. Thus, this qualitized data will be integrated with qualitative data for the overall systematic review.

**Data synthesis and integration.** JBI discusses a number of possibilities for the management of mixed methods data in order to be able to validate or triangulate qualitative and

quantitative findings [33]. Because our approach captures and analyzes data at the same time, and the qualitative and quantitative data will be used to answer the same questions about delays in care, we will utilize a convergent [44] integrated [45] approach for mixed methods systematic reviews. As noted by JBI methodology for mixed methods systematic reviews, we will combine the qualitized data with the qualitative data for analysis so that all extracted data are in qualitative form. The combined extracted data will be synthesized using thematic synthesis [46]. The thematic synthesis will be conducted by two reviewers. Any disagreements will be reconciled by discussion.

First, free codes will be generated based on verbatim findings from the qualitative studies and qualitized data from the quantitative studies. The free codes would then be grouped hierarchically to develop descriptive themes. Analytical themes that answer the research question on factors associated with lung cancer care would then be generated. The analytical themes will be categorized based on Taplin's ecological framework for improving cancer care quality and outcomes– 1) individual, 2) family/social support, 3) provider/team, 4) organization/practice setting, and 5) environment [21]. Results will be presented by textual discussion of the analytical themes as well as the graphical presentation of the ecological framework.

## Discussion

This systematic review will provide a comprehensive synthesis of the evidence on factors that contribute to delays in receiving health services along the lung cancer care continuum. Our multi-level approach will not only explore important individual factors, but it will also examine the impact of the interaction of different factors, and these factors may interact at a single or multi-level. Our theoretical framework promises to shed light on the sometimes confusing array of factors identified in individual studies and collect them to help to be comprehensive and perhaps disambiguate the contribution of single and multiple factors. This ecological framework can also help situate the issue of health disparities in lung cancer outcomes within a broader context of social and environmental factors which may contribute to diagnostic and treatment delays.

Utilization of a mixed method approach allows for the identification of these factors as well as the context in which the factors might influence timeliness of care. By including both qualitative and quantitative data, we will be able to triangulate and provide a more convincing picture of the association of different factors, such as medical mistrust, stigma, and access, to our lung cancer outcomes of interest: delays which affect the morbidity and mortality of diverse populations in the US. Ultimately, findings from this review will inform the design of an MLI to address delays in lung cancer screening for high-risk persons, diagnosis of suspected lung cancer, and treatment of confirmed cases. It will also inform cancer care providers and managers seeking to make quality improvement in the process of care.

### Dissemination of information

Findings will be disseminated at a cancer-focused research meeting with at least one peer-reviewed publication in a health journal.

### Supporting information

**S1 Checklist. PRISMA-P (Preferred Reporting Items for Systematic review and Meta-Analysis Protocols) 2015 checklist: Recommended items to address in a systematic review protocol\*.**
(DOC)

**S1 File. Data extraction spreadsheet.**
(XLSX)

## Author Contributions

**Conceptualization:** Sabina O. Nduaguba.

**Funding acquisition:** Sabina O. Nduaguba, Kimberly M. Kelly.

**Methodology:** Sabina O. Nduaguba, Kimberly M. Kelly.

**Writing – original draft:** Sabina O. Nduaguba.

**Writing – review & editing:** Kimberly M. Kelly.

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
