## [Decision Letter · Decision Letter 0]

9 Jan 2024

PONE-D-23-13627Multilevel factors associated with screening, diagnostic, and treatment delays for lung cancer – A mixed methods systematic review protocolPLOS ONE

Dear Dr. Nduaguba,

Thank you for submitting your manuscript to PLOS ONE. After careful consideration, we feel that it has merit but does not fully meet PLOS ONE’s publication criteria as it currently stands. Therefore, we invite you to submit a revised version of the manuscript that addresses the points raised during the review process.

We look forward to receiving your revised manuscript.

Kind regards,

Caroline Watts, PhD

Academic Editor

PLOS ONE

Journal Requirements:

"Funding for this work was provided by the National Cancer Institute (NCI) Geographic Management of Cancer Health Disparities Program (GMaP). The grant number is: P30 CA177558-10S1. The funder had no role in the development of the protocol."

"Funding for this work was provided to Sabina Nduaguba as a subgrant from the National Cancer Institute (NCI) Geographic Management of Cancer Health Disparities Program (GMaP, http://gmapr1.com/). The grant number is: P30 CA177558-10S1. GMaP played no role in the study design, decision to publish, or preparation of the manuscript."

Reviewers' comments:

Reviewer's Responses to Questions

**Comments to the Author**

1. Does the manuscript provide a valid rationale for the proposed study, with clearly identified and justified research questions?

Reviewer #1: Yes

Reviewer #2: No

2. Is the protocol technically sound and planned in a manner that will lead to a meaningful outcome and allow testing the stated hypotheses?

Reviewer #1: Yes

Reviewer #2: No

3. Is the methodology feasible and described in sufficient detail to allow the work to be replicable?

Reviewer #1: Yes

Reviewer #2: No

4. Have the authors described where all data underlying the findings will be made available when the study is complete?

Reviewer #1: Yes

Reviewer #2: No

5. Is the manuscript presented in an intelligible fashion and written in standard English?

Reviewer #1: Yes

Reviewer #2: Yes

6. Review Comments to the Author

You may also provide optional suggestions and comments to authors that they might find helpful in planning their study.

Reviewer #1: This study developed a mixed methods systematic review protocol to identify multilevel factors associated with delays in lung cancer screening, diagnosis, and treatment. However, there are still some revised suggestions to improve the quality of the paper, as follows.

1. In the Introduction,

①please describe the current status of research on multilevel factors associated with screening, diagnostic, and treatment delays for lung cancer, including quantitative and qualitative studies.

②it is suggested to supplement the detail about the medical mistrust will affect what among providers and lung cancer patients.

2. In the Methods,

①please further explain the quality evaluation criteria, such as what level of quality articles will be excluded?

②it is better to describe how to ensure the accuracy about the transformation process.

3. In the Discussion,

①there is too little content in the discussion section. It is suggested that you add more content to enrich your article.

4. In the Figure,

①the picture is fuzzy, please add the clear figure.

Reviewer #2: Dear Authors,

Congratulations on preparing to undertake an important piece of work on the factors that influence patient attendance to screening, confirmation of diagnosis and receipt of treatment. I agree that this is a timely undertaking given the international interest in the advancement of low dose CT screening and the implementation and evaluation of national lung cancer screening programs. Your comprehensive approach to this topic (i.e., investigating the contexts of screening, diagnosis, treatment) will provide valuable overarching guidance both to clinicians, researchers and policy makers and to your subsequent interventions to improve on delays experienced by patients relative to screening, diagnosis and treatment.

In my review, below, I provide comments and suggestions for improvement of your protocol manuscript. My review is intended in the spirit of good research practice. I do hope my review is helpful to your team.

Major comments:

I have found the conceptual focus of this review quite challenging to understand and, as such, it has been equally challenging to construct my thoughts on how best to write my review. In this view, I choose to start with critique of the section: Data Synthesis and Integration which I believe is at the heart of the conceptual tension that underlies this paper.

Data Synthesis and Integration

This section of the manuscript is perhaps the section that requires most consideration, particularly as the methodological soundness of this section makes or breaks the overall quality of the study. It is not until nearly the end of this manuscript (i.e., pp. 9-10, lines 172-174) that readers are presented with what the authors (I assume) mean by “multilevel factors”. As the paper currently reads and, for those who are experienced in theoretical and/or conceptual approaches that define multilevel influences for use as a generating framework, I have been unclear through the whole of the paper as to the conceptual focus of the study. I have been confused as to why the authors are combining two discrete concepts as their focus, for example, “influences” and “disparities” on timeliness to care. Combining these in the way the authors have done so in the paper introduces conceptual tension. But, after realising that the authors are using the multi-criteria outcomes framework as their conceptual organising framework (critically, this framework is not clearly explicated from the outset; it is not mentioned nor referenced in the Introduction), it becomes somewhat a little clearer as to why the concept of disparities is a focus (per what is stated in the Objectives – “to better understand the multilevel factors influencing disparities in timeliness…”). However, in view of the organising framework used, I argue that it is not that the authors want to “understand the multilevel factors associated with delays in screening, diagnosis, treatment…” because the multilevel factors are defined in the organising framework – patient, clinician, setting, societal. The authors want to understand, within these multilevel planes of reality – i.e., patient, clinician, setting, societal – what factors influence patient attendance to screening, influence patient confirmation of diagnosis and influence patient receipt of treatment. I.E., what patient factors, what clinician factors, what setting factors, what societal factors influence patients relative to these three outcomes of interest?? From this, multi-level delays in these three outcomes of interest can be interpreted. Returning to the concept of “disparities” – is it then just disparities, as a factor, that the authors want to identify/understand? I argue it is conceptually more than this.

Changing focus from the conceptual inconsistencies to the methodological gaps in the paper, on pages 9-10, lines 171-174, the authors state, “Thematic synthesis will be conducted by two reviewers and disagreements reconciled by discussion. The identified themes will be categorized based on the cancer control multilevels for barriers and facilitators from the multi-criteria outcomes framework (i.e. patient, clinician, setting, societal)”. This is the first time that ‘thematic synthesis’ is stated by the authors yet there is no reference or description as to what methodological approach to thematic synthesis they will utilise, for example, classic three-stage approach (coding, generation of descriptive themes, generation and interpretation of analytical themes). Moreover, the authors state that they will use a convergent integrated approach to capturing and analysing data. While I appreciate a convergent approach is to collect and analyse data in parallel and an integrated approach groups studies by their findings (not by their methods) to answer the research question, when/where does the thematic synthesis fit in? In the abstract, the authors state that thematic analysis will be applied to integrate quantitative and qualitative data (thematic analysis is not applied; it is used to generate). In integrated designs, once the studies are grouped, a mixed methods analysis is then undertaken – is this when the authors intend to conduct thematic synthesis? Or will the authors be conducting some form of mixed methods analysis first (per an integrated design) and then completing the analytical process with thematic synthesis as a way of assigning or fitting the findings to the multi-criteria outcomes framework in the Guiding Cancer Control publication?

In this view, I argue that the use of the overarching theoretical organising framework in this paper and the tensioned use of concepts as objectives and outcomes needs a complete rethink.

Additional major comments:

In addition to the above, I provide further thought on the major concepts identified in the paper.

“Multilevel influences”

Following on from my comments above, this concept should be clearly explicated in the Introduction. As mentioned, it is not until reading the section on Data Synthesis and Integration that this concept is made clear – yet, critically, when one reads through the paper, it is conveyed that “multilevel influences” is what the authors aim to identify (but they are already defined in the outcomes criteria framework). Contributing to the unclarity around this concept is the use of different levels of influences cited in the Introduction, for instance, the authors cite a 2009 systematic review that uses individual and health system factors…

“Health disparities”

The authors have included ‘health disparities’ in their search strategy. Notwithstanding that the inclusion of the concept of health disparities in this protocol applies or overlays an assumption that disparities can influence patient attendance to screening, to receiving a diagnosis and/or to receiving treatment (which happens to be in contradiction to the findings of the Slatore et al. study cited by the authors in the Introduction [i.e., see college graduates and increased risk of late stage diagnosis]), this manuscript will thus be best served by fleshing out this concept in the Introduction and making clear its relationship to the proposed review – so that it is unequivocal as to why the authors are including it in addition to identifying factors that influence patient attendance to screening, to receiving a diagnosis and/or to receiving treatment (where influencing factors, if conceptually explicated, can subsume factors or issues that drive disparity).

At times, the authors refer to access. It would be valuable to understand if the authors are intending access to be conceptually relational to health disparities.

“Timeliness”

The concept of “timeliness”, I believe, adds to the conceptual confusion throughout the paper. In wanting to know about the timeliness of patients attending screening, the timeliness of patients receiving a diagnosis, and the timeliness of patients commencing treatment, from a theoretical lens, this approach assumes that patients attend screening, receive a diagnosis and commence treatment – and not all people will attend screening, receive a diagnosis or commence treatment for various reasons. I would encourage the authors to make clear their definitions in the paper regarding “timeliness” – and “delays” – and the perspectives from which these concepts are considered and their relationship to each other and to the study proper.

“Lung cancer care”

The term, lung cancer care, needs definition. Do the authors intend for this to be an umbrella term that subsumes screening, diagnosis and or treatment? Left undefined, it is ambiguous and makes it hard to understand the exact focus of the study. For example, in the Introduction, page 3, line 56: “Disparate times to lung cancer care…” I would argue, times to what component of care exactly?

Title

As the title currently reads, it immediately leaves one wondering if this study centres on understanding health system delays to lung cancer screening, diagnostic, and treatment, but I suspect that the authors aim for this study to focus on delays experienced by patients? I also suspect that upon reconsidering the conceptual inconsistencies and tensions throughout the paper, the title may be rewritten in view of enhancing its clarity.

Additional comments per section

Abstract Objective

Per my comments above, I argue that the objective is not to identify multilevel influences as the multi-levels are defined in the organising framework. I anticipate this stated objective will be crystallised once the concepts and methodological approach are reconsidered.

Further, if the authors could set the context in the Objective, it would enhance clarity in the focus of this study, for example, “To identify…. in the US”).

Abstract Background

“Factors affecting time to lung cancer care….” again, are “disparities” the factors being considered? Is “access” a factor too? I argue that disparities and access, as concepts, do not need to be included at all as it is assumed that these will be explored within each of the planes of reality/planes of influence – i.e., issues of disparity and access at the patient level, at the setting level etc etc

While I understand the need to conduct a systematic review into a topic in which researchers subsequently aim to develop, implement, and evaluate interventions,

we do know a lot about factors – across a multitude of planes – that affect patient experience with lung cancer care. Perhaps the authors can craft the need for this systematic review as an evidence update to previous systematic reviews and studies on this topic? I appreciate this is stated at the end of the Introduction but, if possible to be briefly inserted in the Abstract, it would be valuable.

Introduction

Second para of Introduction, pp 3-4, lines 55-71, I think this para could be broken into two paras. The topics that are currently raised in this para (e.g., factors affecting time to lung cancer care, discrete cohorts, access, mistrust, interaction of factors…) appear “squished” into this para, in a rather “hurried” approach to the construction of the argument. Moreover, I find it interesting that the authors haven’t raised stigma as an influencing factor in this section (perhaps the authors can comment on this?). It would serve this Introduction well if the authors could revisit this second para and spend some time to reconstruct the argument (and explicate the concepts).

Page 4, lines 76-77, the 18 studies that the authors refer to in the Olsson et al. study examined the association between timeliness of lung cancer care and outcomes, not just factors associated with timeliness as the authors’ state. Perhaps the authors might consider tightening their description of this? Further, the authors move on to state “The other two reviews…” but, which reviews are the authors referring to? They reference these two reviews as #29 and #30 but they are not those referred to in the statement on lines 73-74, “Several studies have investigated…” as these ‘several studies’ are referenced as #22, #27, #28. My comments here support my comments above regarding this Introduction ‘feeling rushed’ in its argument construction.

Objectives

“Multilevel”? See my comments above regarding the need for ‘multilevel’ to be defined.

I argue that the stated objective, page 5, lines 94-95, could be clearer. For example, “…associated with delays…” while readers will assume this refers to delays experienced by patients, it could indeed refer to delays experienced by health professionals, or the lung cancer service… this needs to be clearer.

The statement on lines 96-97, “To better understand…”, could be deleted. It confuses an already unclear objective. I suggest that following the first objective statement (which needs reconceptualising), the authors could move straight to, “To achieve this, we propose to address the following aims:”

Stated aims on lines 98-102 need to be clearer per my comments above.

Materials and Methods

Outcomes

If the authors choose to define ‘delay’ based on time intervals, what time intervals are these? For example, are they recommended optimal care pathway time intervals? Or local/contextual time intervals? Similarly, for wait times, will these be recommended/minimum wait times or will they be determined analytically in the evidence collected? Or other guidelined approach?

Informational Sources and Search Strategy

Can the authors explain why they have chosen to search the databases from their inception and not from the date of the last major systematic review undertaken on this topic?

Will any other databases be searched, for example, Embase? Will websites be searched? Unpublished or gray literature?

Eligibility Criteria

Can the authors provide a justification statement for why they have chosen to include both quantitative and qualitative studies, for example, why have they chosen to conduct a systematic mixed study review?

I appreciate letters to the editor, editorials and commentaries will be excluded, but will major reports or similar be considered?

Data Extraction

The authors might consider providing a draft data extraction tool as an appendix, which would enhance this section and the overall transparency and trustworthiness of the planned analytical approach of this study.

Data Transformation

Page 9, line 160, the authors refer to qualitized data in single quote marks. As this is the first time this concept is mentioned, can a reference please be provided? Moreover, the authors explanation of transforming quantitative data into qualitized data needs explication as it is not stated at all.

How do the authors intend to conduct their thematic synthesis of combined qualitized and qualitative data?

It is stated that analysis will be “conducted by two reviewers” – using analysis software? Or researcher driven?

What is meant by the fitting of themes to the categories of the outcomes’ framework?

Additional minor comments

Page 4, line 76 – “…included 18 included…” consider rewording.

Page 4, line 77 – “…over seven countries…” consider rewording.

Check that the DOIs in the reference list all work, as many currently do not.

Overall impression of the study as reported in this manuscript

As mentioned in the outset of my review, I congratulate the authors on preparing to undertake an important piece of work. Due to the tensioned use and misuse of concepts throughout the paper and, including the methodological opaqueness, my overall impression of this paper is to reject it as it currently is. However, with major revision, particularly deep conceptual thought, this body of work will make a valuable contribution to the field of lung cancer care. I note that PLOS ONE encourages authors to publish detailed protocols. I argue that deep thought to inform details is needed. I encourage the authors to continue to improve this paper.

7. PLOS authors have the option to publish the peer review history of their article (what does this mean?). If published, this will include your full peer review and any attached files.

Reviewer #1: No

Reviewer #2: No

---

## [Author Response · Author response to Decision Letter 0]

22 Apr 2024

Response to Reviewer Comments

Reviewer #1: 

This study developed a mixed methods systematic review protocol to identify multilevel factors associated with delays in lung cancer screening, diagnosis, and treatment. However, there are still some revised suggestions to improve the quality of the paper, as follows.

1. In the Introduction,

①please describe the current status of research on multilevel factors associated with screening, diagnostic, and treatment delays for lung cancer, including quantitative and qualitative studies.

Response: We have revised the Introduction Section to include a more robust summary of prior reviews answering similar research questions

②it is suggested to supplement the detail about the medical mistrust will affect what among providers and lung cancer patients.

Response: We have clarified the sentence to read:

“Access to lung cancer care is a health system factor that could further be complicated by mistrust of the health system. In turn, medical mistrust may affect quality of communication between the providers and their patients with lung cancer, ultimately delaying time to receive care” 

2. In the Methods,

①please further explain the quality evaluation criteria, such as what level of quality articles will be excluded?

Response: We have added the following to the Section on Quality Assessment:

“Overall quality rating will be based on the totality of flaws identified. For quantitative studies, a lack of control for confounding will be considered a fatal flaw.”

②it is better to describe how to ensure the accuracy about the transformation process.

Response: We have added an example within the Methods Section

3. In the Discussion,

①there is too little content in the discussion section. It is suggested that you add more content to enrich your article.

Response: We have added additional content to the Discussion Section

4. In the Figure,

①the picture is fuzzy, please add the clear figure.

Response: We have substituted Figure 1 with a clearer figure

Reviewer #2: Dear Authors,

Congratulations on preparing to undertake an important piece of work on the factors that influence patient attendance to screening, confirmation of diagnosis and receipt of treatment. I agree that this is a timely undertaking given the international interest in the advancement of low dose CT screening and the implementation and evaluation of national lung cancer screening programs. Your comprehensive approach to this topic (i.e., investigating the contexts of screening, diagnosis, treatment) will provide valuable overarching guidance both to clinicians, researchers and policy makers and to your subsequent interventions to improve on delays experienced by patients relative to screening, diagnosis and treatment.

In my review, below, I provide comments and suggestions for improvement of your protocol manuscript. My review is intended in the spirit of good research practice. I do hope my review is helpful to your team.

Major comments:

I have found the conceptual focus of this review quite challenging to understand and, as such, it has been equally challenging to construct my thoughts on how best to write my review. In this view, I choose to start with critique of the section: Data Synthesis and Integration which I believe is at the heart of the conceptual tension that underlies this paper.

Data Synthesis and Integration

This section of the manuscript is perhaps the section that requires most consideration, particularly as the methodological soundness of this section makes or breaks the overall quality of the study. It is not until nearly the end of this manuscript (i.e., pp. 9-10, lines 172-174) that readers are presented with what the authors (I assume) mean by “multilevel factors”. As the paper currently reads and, for those who are experienced in theoretical and/or conceptual approaches that define multilevel influences for use as a generating framework, I have been unclear through the whole of the paper as to the conceptual focus of the study. I have been confused as to why the authors are combining two discrete concepts as their focus, for example, “influences” and “disparities” on timeliness to care. Combining these in the way the authors have done so in the paper introduces conceptual tension. But, after realising that the authors are using the multi-criteria outcomes framework as their conceptual organising framework (critically, this framework is not clearly explicated from the outset; it is not mentioned nor referenced in the Introduction), it becomes somewhat a little clearer as to why the concept of disparities is a focus (per what is stated in the Objectives – “to better understand the multilevel factors influencing disparities in timeliness…”). However, in view of the organising framework used, I argue that it is not that the authors want to “understand the multilevel factors associated with delays in screening, diagnosis, treatment…” because the multilevel factors are defined in the organising framework – patient, clinician, setting, societal. The authors want to understand, within these multilevel planes of reality – i.e., patient, clinician, setting, societal – what factors influence patient attendance to screening, influence patient confirmation of diagnosis and influence patient receipt of treatment. I.E., what patient factors, what clinician factors, what setting factors, what societal factors influence patients relative to these three outcomes of interest?? From this, multi-level delays in these three outcomes of interest can be interpreted. Returning to the concept of “disparities” – is it then just disparities, as a factor, that the authors want to identify/understand? I argue it is conceptually more than this.

Response: We have made the following revisions – 1) clarified the study objectives to indicate identification of multilevel factors rather than understanding multilevel factors; 2) updated the framework on which the review is based

Changing focus from the conceptual inconsistencies to the methodological gaps in the paper, on pages 9-10, lines 171-174, the authors state, “Thematic synthesis will be conducted by two reviewers and disagreements reconciled by discussion. The identified themes will be categorized based on the cancer control multilevels for barriers and facilitators from the multi-criteria outcomes framework (i.e. patient, clinician, setting, societal)”. This is the first time that ‘thematic synthesis’ is stated by the authors yet there is no reference or description as to what methodological approach to thematic synthesis they will utilise, for example, classic three-stage approach (coding, generation of descriptive themes, generation and interpretation of analytical themes). Moreover, the authors state that they will use a convergent integrated approach to capturing and analysing data. While I appreciate a convergent approach is to collect and analyse data in parallel and an integrated approach groups studies by their findings (not by their methods) to answer the research question, when/where does the thematic synthesis fit in? In the abstract, the authors state that thematic analysis will be applied to integrate quantitative and qualitative data (thematic analysis is not applied; it is used to generate). In integrated designs, once the studies are grouped, a mixed methods analysis is then undertaken – is this when the authors intend to conduct thematic synthesis? Or will the authors be conducting some form of mixed methods analysis first (per an integrated design) and then completing the analytical process with thematic synthesis as a way of assigning or fitting the findings to the multi-criteria outcomes framework in the Guiding Cancer Control publication?

Response: We are synthesis the data using the three-stage thematic analysis by Thomas (2008). The integration component includes the transformation of quantitative data into qualitized data and integration of the qualitized data with the qualitative data. Our approach is convergent because our review question can be addressed by both quantitative and qualitative studies.

In this view, I argue that the use of the overarching theoretical organising framework in this paper and the tensioned use of concepts as objectives and outcomes needs a complete rethink.

Response: Kindly see our responses to comments above

Additional major comments:

In addition to the above, I provide further thought on the major concepts identified in the paper.

“Multilevel influences”

Following on from my comments above, this concept should be clearly explicated in the Introduction. As mentioned, it is not until reading the section on Data Synthesis and Integration that this concept is made clear – yet, critically, when one reads through the paper, it is conveyed that “multilevel influences” is what the authors aim to identify (but they are already defined in the outcomes criteria framework). Contributing to the unclarity around this concept is the use of different levels of influences cited in the Introduction, for instance, the authors cite a 2009 systematic review that uses individual and health system factors…

Response: We appreciate your comment. Kindly note that the frameworks on which the concepts are structured are generic. Identified multilevel factors are overlayed on the frameworks to organize them into ‘themes’. Our goal is to synthesize the current literature, including qualitative and quantitative studies, on factors associated with delays to lung cancer care. We now specify an ecological framework and briefly describe it in the Introduction Section.

“Health disparities”

The authors have included ‘health disparities’ in their search strategy. Notwithstanding that the inclusion of the concept of health disparities in this protocol applies or overlays an assumption that disparities can influence patient attendance to screening, to receiving a diagnosis and/or to receiving treatment (which happens to be in contradiction to the findings of the Slatore et al. study cited by the authors in the Introduction [i.e., see college graduates and increased risk of late stage diagnosis]), this manuscript will thus be best served by fleshing out this concept in the Introduction and making clear its relationship to the proposed review – so that it is unequivocal as to why the authors are including it in addition to identifying factors that influence patient attendance to screening, to receiving a diagnosis and/or to receiving treatment (where influencing factors, if conceptually explicated, can subsume factors or issues that drive disparity).

Response: In the Introduction Section, we make reference to disparities on singular issues being factors that also influence tiemliness of lung cancer care

“The other two reviews identified investigated singular issues on health disparities and timeliness of lung cancer care – gender[29] and socioeconomic inequality.[30]

At times, the authors refer to access. It would be valuable to understand if the authors are intending access to be conceptually relational to health disparities.

Response: We have clarified this and now refer to timeliness of receipt of cancer care

“Timeliness”

The concept of “timeliness”, I believe, adds to the conceptual confusion throughout the paper. In wanting to know about the timeliness of patients attending screening, the timeliness of patients receiving a diagnosis, and the timeliness of patients commencing treatment, from a theoretical lens, this approach assumes that patients attend screening, receive a diagnosis and commence treatment – and not all people will attend screening, receive a diagnosis or commence treatment for various reasons. I would encourage the authors to make clear their definitions in the paper regarding “timeliness” – and “delays” – and the perspectives from which these concepts are considered and their relationship to each other and to the study proper.

Response: We have clarified our inclusion criteria to include studies on receipt/non-receipt of care as non-receipt of care is also indicative of delays along the cancer care continuum

“Studies that investigate factors associated to whether or not patients receive lung cancer care will also be included as non-receipt of care is also indicative of delays along the cancer care continuum.”

“Lung cancer care”

The term, lung cancer care, needs definition. Do the authors intend for this to be an umbrella term that subsumes screening, diagnosis and or treatment? Left undefined, it is ambiguous and makes it hard to understand the exact focus of the study. For example, in the Introduction, page 3, line 56: “Disparate times to lung cancer care…” I would argue, times to what component of care exactly?

Response: We have refined our outcome definition to indicate delay in receipt of lung cancer care with the specification that this includes delays in lung cancer screening, diagnosis, and treatment.

“Our outcome of interest is delay in receipt of lung cancer care and includesThe three outcomes of interest include: 1) delay in screening for lung cancer; 2) delay in diagnosis of lung cancer; and 3) delay in receipt of lung cancer treatment.”

Title

As the title currently reads, it immediately leaves one wondering if this study centres on understanding health system delays to lung cancer screening, diagnostic, and treatment, but I suspect that the authors aim for this study to focus on delays experienced by patients? I also suspect that upon reconsidering the conceptual inconsistencies and tensions throughout the paper, the title may be rewritten in view of enhancing its clarity.

Response: We have revised the title to read “Multilevel factors associated with delays in screening, diagnosis, and treatment delays for lung cancer – A mixed methods systematic review protocol”

Additional comments per section

Abstract Objective

Per my comments above, I argue that the objective is not to identify multilevel influences as the multi-levels are defined in the organising framework. I anticipate this stated objective will be crystallised once the concepts and methodological approach are reconsidered.

Response: Per our comment above, the frameworks on which the concepts are structured are generic. Identified multilevel factors are overlayed on the frameworks to organize them into ‘themes’. 

Further, if the authors could set the context in the Objective, it would enhance clarity in the focus of this study, for example, “To identify…. in the US”).

Response: We have revised the sub-aims as suggested

Abstract Background

“Factors affecting time to lung cancer care….” again, are “disparities” the factors being considered? Is “access” a factor too? I argue that disparities and access, as concepts, do not need to be included at all as it is assumed that these will be explored within each of the planes of reality/planes of influence – i.e., issues of disparity and access at the patient level, at the setting level etc etc

Response: The abstract background is consistent with the point made in your comment. We do not make any changes at this time

While I understand the need to conduct a systematic review into a topic in which researchers subsequently aim to develop, implement, and evaluate interventions,

we do know a lot about factors – across a multitude of planes – that affect patient experience with lung cancer care. Perhaps the authors can craft the need for this systematic review as an evidence update to previous systematic reviews and studies on this topic? I appreciate this is stated at the end of the Introduction but, if possible to be briefly inserted in the Abstract, it would be valuable.

Response: We have updated the Abstract Objective to read: “We aimed to update the literature on systematic reviews and identify multilevel factors associated with delays in lung cancer screening, diagnosis, and treatment”

Introduction

Second para of Introduction, pp 3-4, lines 55-71, I think this para could be broken into two paras. The topics that are currently raised in this para (e.g., factors affecting time to lung cancer care, discrete cohorts, access, mistrust, interaction of factors…) appear “squished” into this para,

---

## [Decision Letter · Decision Letter 1]

11 Jul 2024

PONE-D-23-13627R1Multilevel factors associated with delays in screening, diagnosis, and treatment for lung cancer – A mixed methods systematic review protocolPLOS ONE

Dear Dr. Nduaguba,

Thank you for submitting your manuscript to PLOS ONE. After careful consideration, we feel that it has merit but does not fully meet PLOS ONE’s publication criteria as it currently stands. Therefore, we invite you to submit a revised version of the manuscript. I believe you have addressed the comments from the reviewers but ask that you review the Methods and in particular your use of the word fatal in the sentence, "For quantitative studies, a lack of control for confounding will be considered a fatal flaw” and rewrite this sentence.

We look forward to receiving your revised manuscript.

Kind regards,

Caroline Watts, PhD

Academic Editor

PLOS ONE

Journal Requirements:

Additional Editor Comments:

I believe you have addressed the comments from the reviewers but ask that you review your use of the word fatal in the sentence, "For quantitative studies, a lack of control for confounding will be considered a fatal flaw” and rewrite this sentence.

Reviewers' comments:

Reviewer's Responses to Questions

**Comments to the Author**

1. Does the manuscript provide a valid rationale for the proposed study, with clearly identified and justified research questions?

Reviewer #3: Yes

2. Is the protocol technically sound and planned in a manner that will lead to a meaningful outcome and allow testing the stated hypotheses?

Reviewer #3: Yes

3. Is the methodology feasible and described in sufficient detail to allow the work to be replicable?

Reviewer #3: Yes

4. Have the authors described where all data underlying the findings will be made available when the study is complete?

Reviewer #3: Yes

5. Is the manuscript presented in an intelligible fashion and written in standard English?

Reviewer #3: Yes

6. Review Comments to the Author

You may also provide optional suggestions and comments to authors that they might find helpful in planning their study.

Reviewer #3: Thank you for the opportunity to review this manuscript. The authors have considerably responded to the reviewer’s comments. However, I would like to make some minor suggestions to improve the manuscript.

Objectives

Page 6, lines 93…The authors should go through their objective to correct the sentence or add (is) to “the objective of this systematic review (is) to identify multilevel factors associated with delays experienced by patients along the lung cancer care continuum from screening to diagnosis and treatment in the US”.

Outcomes

Page 6, line 110…” Delay is defined based…” (be) should be removed from the sentence.

Eligibility criteria: page 9, lines 132-133. What happens where both reviewers do not reach a consensus?. Will the authors involve a third reviewer to resolve their differences or how do they intend to resolve their discrepancies when the both of them disagree.

I would recommend accepting the manuscript.

7. PLOS authors have the option to publish the peer review history of their article (what does this mean?). If published, this will include your full peer review and any attached files.

Reviewer #3: **Yes: **Ugochinyere Ijeoma Nwagbara

---

## [Author Response · Author response to Decision Letter 1]

29 Jul 2024

Additional Editor Comments:

I believe you have addressed the comments from the reviewers but ask that you review your use of the word fatal in the sentence, "For quantitative studies, a lack of control for confounding will be considered a fatal flaw” and rewrite this sentence.

Response: We have deleted the sentence, as given the comment, deletion does not detract from the methodology.

Reviewer #3: Thank you for the opportunity to review this manuscript. The authors have considerably responded to the reviewer’s comments. However, I would like to make some minor suggestions to improve the manuscript.

Response: We have addressed the comments. See below.

Objectives

Page 6, lines 93…The authors should go through their objective to correct the sentence or add (is) to “the objective of this systematic review (is) to identify multilevel factors associated with delays experienced by patients along the lung cancer care continuum from screening to diagnosis and treatment in the US”

Response: We have applied the suggestion.

Outcomes

Page 6, line 110…” Delay is defined based…” (be) should be removed from the sentence.

Response: We have applied the suggestion

Eligibility criteria: page 9, lines 132-133. What happens where both reviewers do not reach a consensus?. Will the authors involve a third reviewer to resolve their differences or how do they intend to resolve their discrepancies when the both of them disagree.

Response: To address this comment, we added the following sentence: “In the event that consensus is not reached, input will be sought from a third reviewer to resolve the discrepancy.”

---

## [Editor Report · Decision Letter 2]

7 Aug 2024

Multilevel factors associated with delays in screening, diagnosis, and treatment for lung cancer – A mixed methods systematic review protocol

PONE-D-23-13627R2

Dear Dr. Nduaguba,

We’re pleased to inform you that your manuscript has been judged scientifically suitable for publication and will be formally accepted for publication once it meets all outstanding technical requirements.

Kind regards,

Caroline Watts, PhD

Academic Editor

PLOS ONE

---

## [Editor Report · Acceptance letter]

15 Aug 2024

PONE-D-23-13627R2 

PLOS ONE

Dear Dr. Nduaguba, 

I'm pleased to inform you that your manuscript has been deemed suitable for publication in PLOS ONE. Congratulations! Your manuscript is now being handed over to our production team.

Kind regards, 

on behalf of

Dr. Caroline Watts 

Academic Editor

PLOS ONE